

# Analysis of influencing factors of serum total protein and serum calcium content in plasma donors

Bin Liu[1,*], Demei Dong[2,*], Zongkui Wang[1], Yang Gao[2], Ding Yu[3], Shengliang Ye[1], Xi Du[1], Li Ma[1], Haijun Cao[1], Fengjuan Liu[1], Rong Zhang[1] and Changqing Li[1]

[1] Institute of Blood Transfusion, Chinese Academy of Medical Sciences & Peking Union Medical College, Chengdu, China
[2] Beijing Tiantan Biological Products Co., Ltd, Chengdu, China
[3] Rongsheng Pharmaceuticals Co., Ltd, Chengdu, China
[*] These authors contributed equally to this work.

Corresponding authors
Rong Zhang,
rong.zhang@ibt.pumc.edu.cn,
kylie2009@foxmail.com
Changqing Li,
changqing.li@ibt.pumc.edu.cn,
lcq@ibt.pumc.edu.cn

## ABSTRACT

**Background and objectives.** The adverse effects of plasma donation on the body has lowered the odds of donation. The aim of this study was to investigate the prevalence of abnormal serum calcium and total serum protein related to plasma donation, identify the influencing factors, and come up with suggestions to make plasma donation safer.
**Methods.** Donors from 10 plasmapheresis centers in five provinces of China participated in this study. Serum samples were collected before donation. Serum calcium was measured by arsenazo III colorimetry, and the biuret method was used for total serum protein assay. An automatic biochemical analyzer was used to conduct serum calcium and total serum protein tests.
**Results.** The mean serum calcium was $2.3 \pm 0.15$ mmol/L and total serum protein was $67.75 \pm 6.02$ g/L. The proportions of plasma donors whose serum calcium and total serum protein were lower than normal were 20.55% (815/3,966) and 27.99% (1,111/3,969), respectively. There were significant differences in mean serum calcium and total serum protein of plasma donors with different plasma donation frequencies, gender, age, regions, and body mass index (BMI), (all $p < 0.05$). Logistic regression analysis revealed that donation frequencies, age, BMI and regions were significantly associated with a higher risk of low serum calcium level, and donation frequencies, gender, age and regions were significant determinants factors of odds of abnormal total serum protein.
**Conclusions.** Donation frequencies, gender, age, regions, and BMI showed different effects on serum calcium and total serum protein. More attention should be paid to the age, donation frequency and region of plasma donors to reduce the probability of low serum calcium and low total serum protein.

## INTRODUCTION

As important medicines for clinical use, blood and blood components save millions of lives each year and are included in the Model List of Essential Medicines of the World Health Organization (WHO) (*Organization, 2021*). Plasma-derived medicinal products (PDMPs), such as albumin, coagulation factors and immunoglobulins are prepared from human plasma. They are used to prevent and treat a variety of life-threatening diseases (*Grazzini, Mannucci & Oleari, 2013*). The statistical results of WHO in 2020 showed that only 56 of 171 reporting countries produced PDMPs through the fractionation of plasma, and 91 countries reported that all PDMPs were imported (*Organization, 2020*). The risk for PDMPs shortages, as well as increasing demand, may result in depriving patients of essential medicinal products (*Tiberghien, 2021*).

Source plasma (SP) is an important raw material for the production of PDMPs and is used exclusively for further manufacturing into final therapies (fractionation). Recovered plasma is collected through whole blood donation in which plasma is separated from its cellular components. Recovered plasma may be used for fractionation. Most SP (85%–90%) is collected by apheresis from donors and a small fraction (10–15%) is contributed by recovered plasma (*Hartmann & Klein, 2020*). In China, all SP is obtained from apheresis plasma. Donation serves the demand for plasma but there are concerns among potential donors about the impact of blood loss on physical health. Concerns about the health impact of plasma donation limit the number of plasma donations (*Thorpe et al., 2020*).

It is reported that donors perceive both positive and negative effects of blood donation (*Teglkamp et al., 2020*). The most often reported positive symptoms are: alleviated headache, feeling lighter, and less tiredness (*Teglkamp et al., 2020*; *Hinrichs et al., 2008*; *Sojka & Sojka, 2003*; *Van DenHurk et al., 2017*), and the most common negative effects are iron deficiency, vasovagal reactions and citrate-related events (*Amrein et al., 2012*). Most researches on adverse donor reactions focused on whole blood donation, little attention paid to adverse reactions of plasma donation (*Amrein et al., 2012*; *Crocco et al., 2009*; *Almutairi et al., 2017*; *Hu et al., 2019*; *Locks et al., 2019*; *Newman, 2004a*; *Newman, 2004b*; *Orru et al., 2021*; *Prakash et al., 2020*). In order to further understand the impact of plasma donation on donors and to come up with suggestions which can make voluntary plasma donation safer, we investigated the changes of serum calcium and total serum protein in different types of plasma donors. The factors that may affect the changes of serum total protein and serum calcium were also discussed.

## MATERIAL AND METHODS

### Sample collection

This was a cross-sectional, multicenter study. A total of 4,000 subjects (2,000 male and 2,000 female) were recruited for this study. All participants were healthy plasma donors who donated during Jun. 25, 2021 to Sep.4, 2021, and came from 10 plasmapheresis centers in five provinces (Sichuan, Jiangxi, Hubei, Shandong, and Ningxia) of China. The inclusion criteria were that all participators were 18-60 years, healthy, and unrelated. People who had history of thrombus or hemorrhage, usage of oral anticoagulation therapy,

pregnancy, HIV infection, hepatic disease, diabetes, renal insufficiency, severe vitamin D deficiency, hyperparathyroidism, chronic inflammatory syndrome, *etc.* were excluded from this research.

Five microliter venous blood of each participant was collected in a sterile tube without anticoagulant before plasmapheresis. After coagulation, each sample was centrifuged at 3,000 g for 10 min. The serum was separated from each sample and stored at−70 °C in two aliquots separately until transported to the institute of blood transfusion (IBT) by dry ice. When each sample reached the laboratory of IBT, one aliquot was immediately used for serum calcium detection, and the other aliquot was used for total serum protein detection.

After excluding incomplete information, duplicate and data drift samples, 3,966 and 3,969 subjects were available in the statistical analysis of serum calcium and total serum protein, respectively. The demographic information of participants is shown in Table 1.

## Laboratory assays

Serum calcium and total serum protein were performed using an automatic biochemical analyzer (Beckman Coulter AU5800; Beckman Coulter Inc., La Brea CA, USA) according to the manufacture's protocols by arsenazo III colorimetry and biuret method, respectively. The reagents were purchased from Beckman Coulter, Inc. (La Brea CA, USA). The reference range of serum calcium was 2.20−2.65 mmol/L, and the reference range of total serum protein was 65–85 g/L.

## Statistical analysis

The Kolmogorov–Smirnov test was used for the normal distribution of all data. All values were expressed as means $\pm$ standard deviation (SD). Multi-group comparisons (different donation frequencies, age categories, regions, blood type groups, and BMI) were conducted by one-way ANOVA followed by LSD *post hoc* test. The effects of gender on serum calcium and total serum protein were accomplished by using two-tailed unpaired Student's *t* -tests. Poisson analysis was used to determine the associations between donation frequencies and serum calcium, donation frequencies and total serum protein, age and serum calcium, age and total serum protein, BMI and serum calcium, BMI and total serum protein. A correlation coefficients of <0.10, 0.10−0.29, 0.30−0.49, and ⩾0.50 were considered negligible, small, moderate, and large correlation, respectively. A 95% confidence intervals (CI) was used and a *p*-value <0.05 was considered significant. Furthermore, binary logistic regression analyses were used to investigate the effect of different influencing factors on odds of abnormal serum calcium and total serum protein (abnormal serum calcium: value was lower than 2.20 mmol/L; abnormal total serum protein: value was lower than 65 g/L). The strengths of the relationships were expressed as odds ratios (ORs) with corresponding 95% CIs, and *p* values were calculated for the corresponding results in the logistic regression. According to the Working Group on Obesity in China (WGOC) (*Zhou, 2002*), 24 kg/m$^2$ ≤BMI <28 kg/m$^2$ and BMI ≥28 kg/m$^2$ were considered as overweight and obesity respectively. Statistical analysis was performed using SPSS statistics software version 22.0 (SPSS Inc., Chicago, USA).

| Table 1 Characteristics of the participants. | | | |
|---|---|---|---|
| | | Serum calcium study ($n = 3{,}966$) | Total serum protein study ($n = 3{,}969$) |
| Age (years) | | $40.50 \pm 11.53$ | $40.49 \pm 11.53$ |
| Gender | Male | 1,986 (50.1%) | 1,989 (50.1%) |
| | Female | 1,980 (49.9%) | 1,980 (49.9%) |
| Blood type | O | 1,286 (32.4%) | 1,288 (32.5%) |
| | A | 1,224 (30.9%) | 1,225 (30.9%) |
| | B | 1,119 (28.2%) | 1,119 (28.2%) |
| | AB | 337 (8.5%) | 337 (8.5%) |
| Donation frequencies[a] | 0 | 1,000 (25.2%) | 1,000 (25.2%) |
| | 1–6 | 988 (24.9%) | 991 (25.0%) |
| | 7–11 | 978 (24.7%) | 978 (24.6%) |
| | 12–27 | 1,000 (25.2%) | 1,000 (25.2%) |
| Regions | Sichuan | 799 (20.1%) | 799 (20.1%) |
| | Jiangxi | 799 (20.1%) | 799 (20.1%) |
| | Hubei | 798 (20.1%) | 798 (20.1%) |
| | Shandong | 772 (19.5%) | 775 (19.5%) |
| | Ningxia | 798 (20.1%) | 798 (20.1%) |
| BMI (kg/m$^2$)[b] | <18.5 | 86 (2.4%) | 85 (2.4%) |
| | 18.5–23.9 | 1,361 (38.2%) | 1,363 (38.2%) |
| | 24.0–27.9 | 1,404 (39.4%) | 1,406 (39.4%) |
| | $\geq 28$ | 716 (20.1%) | 716 (20.1%) |

**Notes.**

BMI, Body mass index.

Data are shown as n (%), mean ± standard deviation.

[a] Number of donors who made a certain number of donations in the year prior to this sample collection.

[b] Excluding incomplete information and duplicate samples, 3,567 and 3,570 subjects were respectively available to serum calcium and total serum protein research in this study.

## Ethics statement

This study was approved by the Ethics Committee of the Institute of IBT, CAMS&PUMC (2021029). Informed consent was obtained from the individual participants according to the Declaration of Helsinki.

## RESULTS

As Table 2 showed, the mean level of serum calcium ($2.3 \pm 0.15$ mmol/L) and total serum protein ($67.75 \pm 6.02$ g/L) of all participants were in normal value range ($2.20-2.65$ mmol/L and 65–85 g/L). Approximately 20.55% (815/3,966) plasma donors' serum calcium was lower than normal value range. The proportion of donors whose total serum protein level was lower than normal value range was 27.99% (1,111/3,969).

### Effects of age, gender and blood type on serum calcium

As shown in Fig. 1A, serum calcium level of the 18–29 year-old group was higher than other three different age groups ($2.35 \pm 0.16$ mmol/L $vs.$ $2.29 \pm 0.15$ mmol/L $vs.$ $2.28 \pm 0.14$ mmol/L $vs.$ $2.30 \pm 0.13$ mmol/L; all $p < 0.001$). Significant difference was also noted

**Table 2** Results of serum calcium and total serum protein in donors, separately for normal and abnormal[*].

| Variables | | Abnormal group | Normal group | Total |
|---|---|---|---|---|
| Serum calcium | n (n, %) | 815, 20.55% | 3,151, 79.45% | 3,966 |
| | mean (mmol/L) | 2.07 ± 0.93 | 2.36 ± 0.93[***] | 2.30 ± 0.15 |
| Total serum protein | n (n, %) | 1,111, 27.99% | 2,858, 72.01% | 3,969 |
| | mean (g/L) | 60.26 ± 4.01 | 70.67 ± 3.72[***] | 67.75 ± 6.02 |

**Notes.**

[*]Data are reported as mean (±SD) or number (%).

[***]$p < 0.001$ (the comparison between abnormal group and normal group was conducted using two-tailed unpaired Student's $t$-tests).

between 40-49 and 50–60 year-old groups (2.28 ± 0.14 mmol/L *vs.* 2.30 ± 0.13 mmol/L; $p = 0.001$). According with these results, the 18–29 year-old group had the lowest abnormal rate of serum calcium level (proportion of people whose serum calcium level was lower than normal value range) among the four different age groups (Table 3, 16.6% *vs.* 21.5% *vs.* 24.3% *vs.* 18.6%). A small and negative correlation (r = −0.137, $p < 0.001$) between serum calcium levels and age was found (Fig. 2A). Furthermore, there was a clear distinction between genders, where serum calcium levels were observably higher in male than in female (2.32 ± 0.15 mmol/L *vs.* 2.29 ± 0.14 mmol/L; $p < 0.001$) (Fig. 1A). And the abnormal rate of serum calcium level of male was lower than female (Table 3, 19.6% *vs.* 21.2%). Whereas ABO blood type showed no effect on serum calcium level (Fig. 1A).

## Effects of age, gender and blood type on total serum protein

Similar to the results of serum calcium, among the four different age groups, the 18-29 year old group had the highest total serum protein level (69.32 ± 6.40 g/L *vs.* 67.48 ± 5.98 g/L *vs.* 67.30 ± 5.93 g/L *vs.* 67.36 ± 5.70 g/L; all $p < 0.001$) (Fig. 1B). Consistent with these results, the 18-29 year old group had the lowest abnormal rate of total serum protein level (proportion of people whose total serum protein level was lower than normal value range) among the four different age groups (Table 3, 22.6% *vs.* 29.2% *vs.* 30.5% *vs.* 28.1%). Age had a small negative correlation with total serum protein levels (r = −0.120, $p < 0.001$) (Fig. 2B). Moreover, ABO blood type showed no effect on total serum protein level, nevertheless total serum protein levels of female were observably higher than that of male (67.95 ± 6.09 g/L *vs.* 67.56 ± 5.95 g/L; $p = 0.037$) (Fig. 1B).

## Effects of donation frequencies, regions and BMI on serum calcium

The level of serum calcium was significantly higher in new donors (donation frequency was 0 times) than in 1–6, 7–11 and 12–27 donation times groups (2.33 ± 0.13 mmol/L *vs.* 2.27 ± 0.16 mmol/L *vs.* 2.30 ± 0.15 mmol/L *vs.* 2.31 ± 0.13 mmol/L; all $p < 0.01$) (Fig. 3A). The new donors had lower abnormal rate of serum calcium level than other three groups (Table 2, 14.5% *vs.* 29.5% *vs.* 20.4% *vs.* 18.0%). In the five different regions, serum calcium level of Shandong participants was lower than other four groups (2.20 ± 0.10 mmol/L *vs.* 2.36 ± 0.10 mmol/L *vs.* 2.29 ± 0.13 mmol/L *vs.* 2.27 ± 0.15 mmol/L *vs.* 2.38 ± 0.10 mmol/L; all $p < 0.001$) (Fig. 3A). Consistently, the abnormal rate of serum calcium level of Shandong participants was higher than other four groups (Table 3, 47.0% *vs.* 3.5% *vs.*

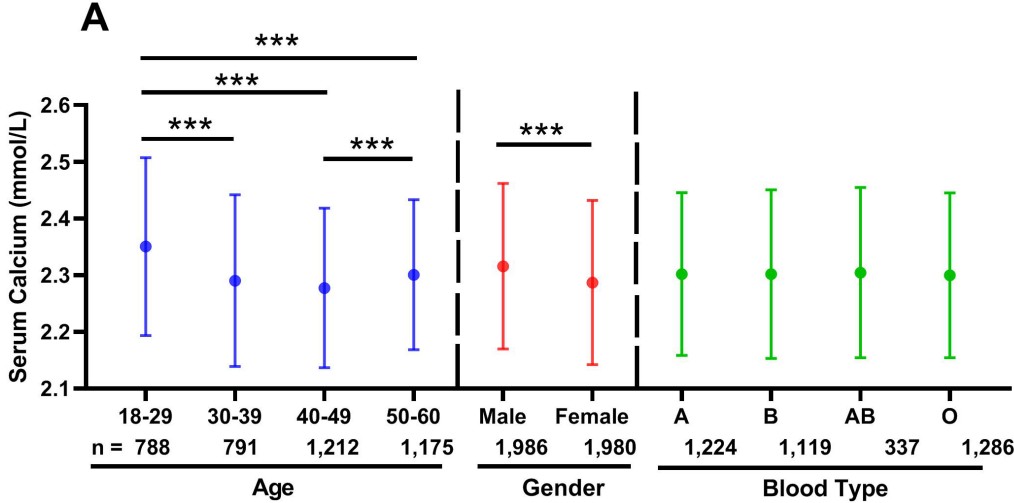

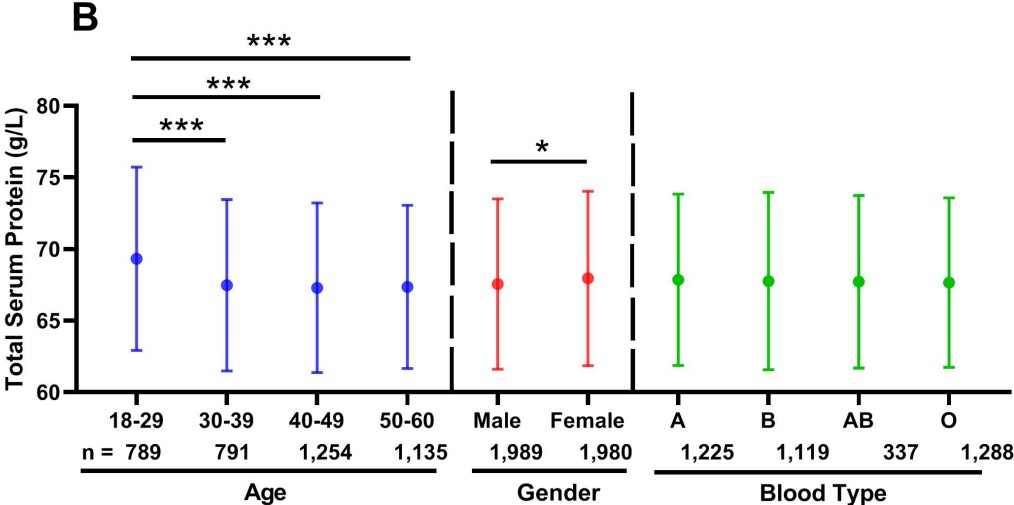

**Figure 1** **Effects of age, gender and blood type on distribution of serum calcium and total serum protein levels.** The solid circle (●) showing the mean values. The effects of age and blood type were calculated using one-way ANOVA followed by LSD *post hoc* test, whereas the influence of gender was calculated using two-tailed unpaired Student's t -tests. $*p < 0.05$; $**p < 0.01$; $***p < 0.001$. (A) Serum calcium; (B) total serum protein.

22.0% *vs.* 29.0% *vs.* 2.1%). In addition, serum calcium level of low BMI ($\leq$18.4) group was higher than other three groups ($2.37 \pm 0.14$ mmol/L *vs.* $2.30 \pm 0.14$ mmol/L *vs.* $2.29 \pm 0.15$ mmol/L *vs.* $2.29 \pm 0.15$ mmol/L; all $p < 0.001$) (Fig. 3A). No correlation between donation frequencies and serum calcium was found, and the correlation between BMI and serum calcium was negligible ($r = -0.056$, $p = 0.0009$) (Fig. 2C).

**Table 3  The proportion of people whose detection value is lower than the normal value range in different groups.**

|  |  | Serum calcium | Total serum protein |
|---|---|---|---|
| Age (years) | 18–29 | 16.6% (131/788) | 22.6% (178/789) |
|  | 30–39 | 21.5% (170/791) | 29.2% (231/791) |
|  | 40–49 | 24.3% (295/1212) | 30.5% (383/1254) |
|  | 50–60 | 18.6% (219/1175) | 28.1% (319/1135) |
| Gender | Male | 19.6% (389/1986) | 28.8% (573/1989) |
|  | Female | 21.2% (426/1980) | 27.2% (538/1980) |
| Blood type | O | 21.2% (273/1286) | 27.3% (352/1288) |
|  | A | 20.2% (247/1224) | 27.9% (342/1225) |
|  | B | 20.3% (227/1119) | 28.6% (320/1119) |
|  | AB | 20.2% (68/337) | 28.8% (97/337) |
| Donation frequencies[a] | 0 | 14.5% (145/1000) | 16.1% (161/1000) |
|  | 1–6 | 29.5% (291/988) | 36.8% (365/991) |
|  | 7–11 | 20.4% (199/978) | 29.5% (288/978) |
|  | 12–27 | 18.0% (180/1000) | 29.7% (297/1000) |
| Regions | Sichuan | 3.5% (28/799) | 13.6% (109/799) |
|  | Jiangxi | 22.0% (176/799) | 31.9% (255/799) |
|  | Hubei | 29.0% (231/798) | 33.7% (269/798) |
|  | Shandong | 47.0% (363/772) | 52.5% (407/775) |
|  | Ningxia | 2.1% (17/798) | 8.9% (71/798) |
| BMI (kg/m$^2$)[b] | <18.5 | 14.0% (12/86) | 21.2% (18/85) |
|  | 18.5–23.9 | 21.3% (290/1361) | 30.4% (414/1363) |
|  | 24.0–27.9 | 23.3% (327/1404) | 29.5% (414/1406) |
|  | ≥ 28 | 23.2% (166/716) | 28.8% (206/716) |

Notes.

BMI, Body mass index.

The normal values of serum calcium and total serum protein were 2.20–2.65 mmol/L and 65–85 g/L respectively.

[a]Number of donors who made a certain number of donations in the year prior to this sample collection.

[b]Excluding incomplete information and duplicate samples, 3,567 and 3,570 subjects were respectively available to serum calcium and total serum protein research in this study.

## Effects of donation frequencies, regions and BMI on total serum protein

The effects of donation frequencies, regions and BMI on total serum protein were similar to those of serum calcium. As Fig. 3B showed, the total serum protein level of new donors was the highest one of the four different donation frequencies groups (69.89 ± 5.25 g/L vs. 66.47 ± 6.63 g/L vs. 67.39 ± 6.40 g/L vs. 67.25 ± 5.14 g/L, $p < 0.001$). Shandong participants' total serum protein level was lower than other four regions groups (64.23 ± 7.32 g/L vs.69.53 ± 4.57 g/L vs. 67.31 ± 5.24 g/L vs. 67.03 ± 6.27 g/L vs. 70.57 ± 4.13 g/L; all $p < 0.001$). And total serum protein level of low BMI group was higher than other three groups (69.87 ± 6.08 g/L vs. 67.58 ± 6.10 g/L vs. 67.47 ± 6.12 g/L vs. 67.48 ± 6.10 g/L; $p = 0.001$, respectively).

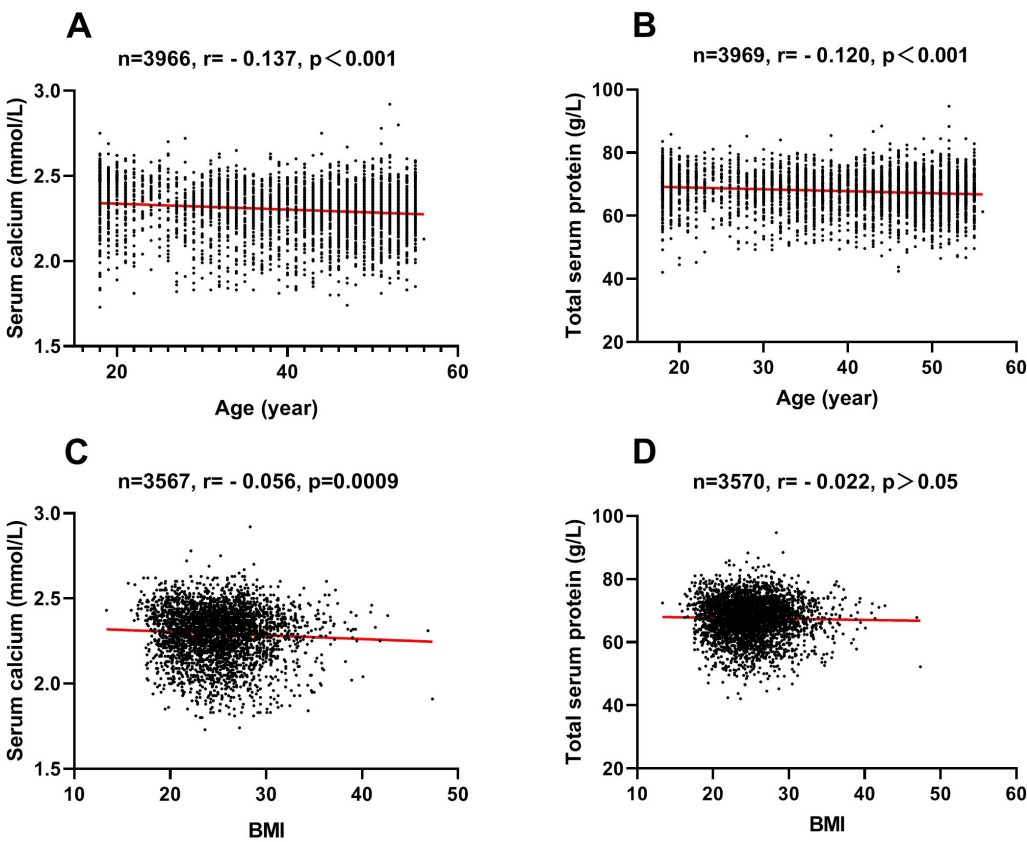

**Figure 2** **Associations of serum calcium and age, total serum protein and age, serum calcium and BMI, and total serum protein and BMI.** Bivariate correlation analysis was used and the diagonal lines indicate linear regression. (A) association of serum calcium and age; (B) association of total serum protein and age; (C) association of serum calcium and BMI; (D) association of total serum protein and BMI. Serum calcium and total serum protein were small negative association with age.

## Results of regression analysis between influencing factors and odds of abnormal serum calcium/total serum protein

The results of binary logistic regression analysis indicated that donation frequencies, age, BMI and regions were significantly associated with a higher risk of low serum calcium level. The OR of repeat donors (donation frequencies were 1-6 times and 7–11 times) were 3.004 times and 1.610 times higher than that of new donors, respectively. The OR of older participants (aged 30–39 years, 40–49 years, and 50–60 years) were 1.392 times, 2.200 times and 1.748 times higher than that of younger participants (aged 18–29 years). The OR of donors whose BMI was 24.0–27.9 was 1.872 times higher than that of donors whose BMI was less than 18.5. The OR of Jiangxi, Hubei, Sichuan and Ningxia participants were decreased by 72.8% (95% CI [0.216–0.342], $p < 0.001$), 61.4% (95% CI [0.309–0.482], $p < 0.001$), 97.0% (95% CI [0.020–0.046], $p < 0.001$) and 97.8% (95% CI [0.013–0.036], $p < 0.001$) compared to Shandong participants, respectively (Table 4).

In the binary logistic regression model, donation frequencies, gender, age and regions were significant determinants factors of odds of abnormal total serum protein. The

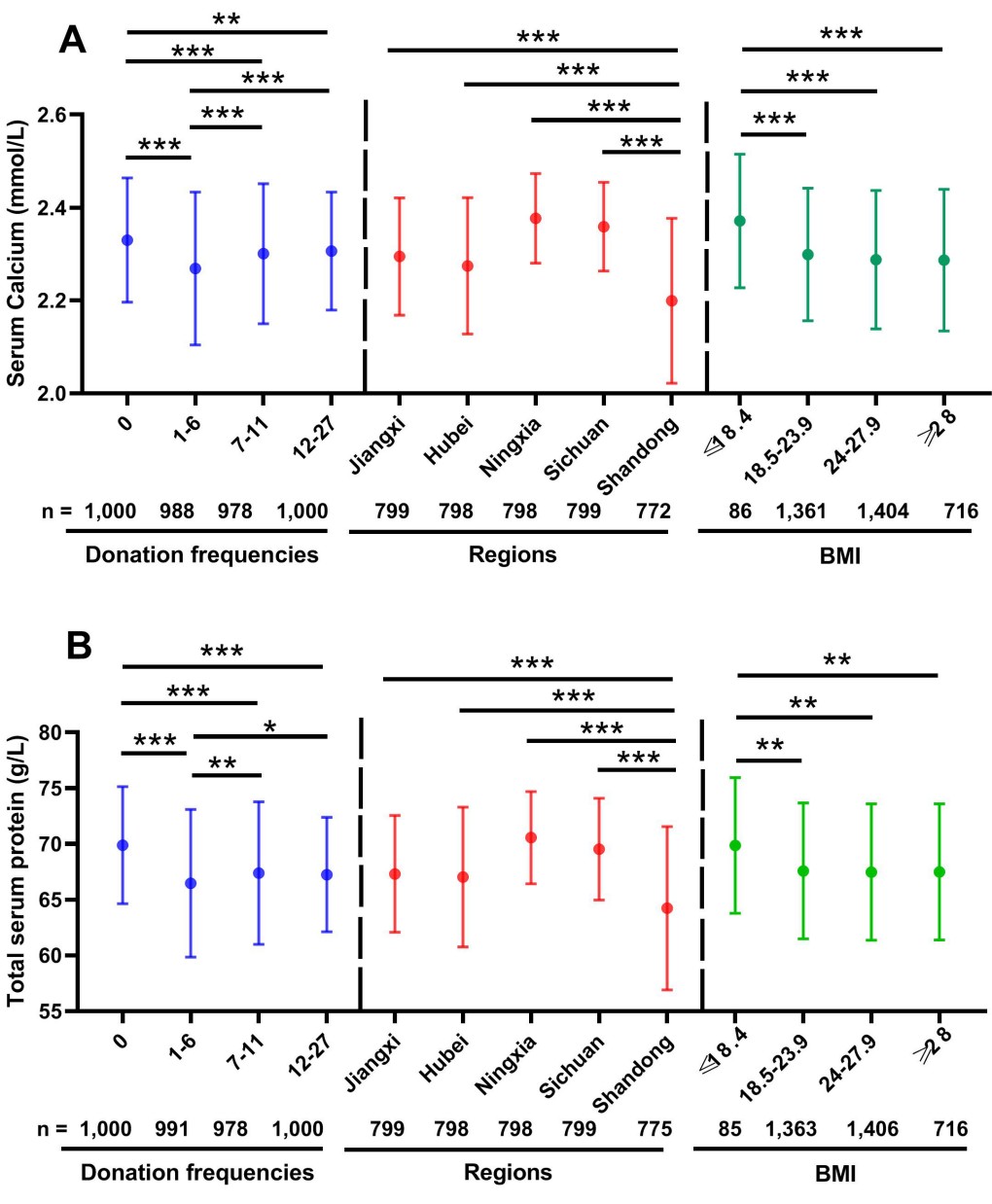

**Figure 3** **Effects of donation frequencies, regions and BMI on distribution of serum calcium and total serum protein levels.** The solid circle (●) showing the mean values. The effects of donation frequencies, regions and BMI were calculated using one-way ANOVA followed by LSD *post hoc* test. $*p < 0.05$; $**p < 0.01$; $***p < 0.001$. (A) Serum calcium; (B) total serum protein.

abnormal total serum protein odds of donors whose donation frequencies were 1–6 times, 7–11 times and 12–27 times were 3.494 times (95% CI [2.781–4.390], $p < 0.001$), 2.366 times (95% CI [1.875–2.984], $p < 0.001$), and 2.179 times (95% CI [1.721–2.760], $p < 0.001$) higher than that of new donors, respectively. The OR of female was decreased by 17.1% (95% CI [0.709–0.969], $p = 0.019$) compared to male. Donors aged 30-39 years, 40-49 years, and 50-60 years had 1.436 times (95% CI [1.115–1.850], $p = 0.005$), 1.889 times

**Table 4  Binary logistic regression analysis between influencing factors and odds of abnormal serum calcium (lower than normal value range).**

| | B | SE (B) | Wald $\chi^2$ | $p$-value | OR | OR (95% CI) |
|---|---|---|---|---|---|---|
| **Donation frequencies** | | | | | | |
| 0 | 1.00 (referent) | | | | | |
| 1–6 | 1.100 | 0.127 | 74.546 | 0.000* | 3.004 | 2.340–3.855 |
| 7–11 | 0.476 | 0.132 | 13.081 | 0.000* | 1.610 | 1.244–2.084 |
| 12–27 | 0.159 | 0.137 | 1.358 | 0.244 | 1.173 | 0.897–1.533 |
| **Gender** | | | | | | |
| Male | 1.00 (referent) | | | | | |
| Female | 0.029 | 0.092 | 0.099 | 0.753 | 1.029 | 0.860–1.231 |
| **Age (years)** | | | | | | |
| 18–29 | 1.00 (referent) | | | | | |
| 30–39 | 0.331 | 0.146 | 5.103 | 0.024* | 1.392 | 1.045–1.855 |
| 40–49 | 0.788 | 0.138 | 32.473 | 0.000* | 2.200 | 1.677–2.885 |
| 50–60 | 0.558 | 0.147 | 14.469 | 0.000* | 1.748 | 1.311–2.330 |
| **Blood type** | | | | | | |
| A | 1.00 (referent) | | | | | |
| B | −0.110 | 0.117 | 0.885 | 0.347 | 0.896 | 0.713–1.126 |
| AB | −0.172 | 0.171 | 1.015 | 0.314 | 0.842 | 0.602–1.177 |
| O | 0.040 | 0.111 | 0.132 | 0.717 | 1.041 | 0.837–1.295 |
| **BMI** | | | | | | |
| <18.5 | 1.00 (referent) | | | | | |
| 18.5–23.9 | 0.513 | 0.318 | 2.597 | 0.107 | 1.670 | 0.895–3.115 |
| 24.0–27.9 | 0.627 | 0.318 | 3.901 | 0.048* | 1.872 | 1.005–3.489 |
| ≥28 | 0.621 | 0.324 | 3.686 | 0.055 | 1.861 | 0.987–3.509 |
| **Regions** | | | | | | |
| Shandong | 1.00 (referent) | | | | | |
| Jiangxi | −1.304 | 0.118 | 122.388 | 0.000* | 0.272 | 0.216–0.342 |
| Hubei | −0.951 | 0.113 | 70.327 | 0.000* | 0.386 | 0.309–0.482 |
| Sichuan | −3.501 | 0.212 | 271.501 | 0.000* | 0.030 | 0.020–0.046 |
| Ningxia | −3.825 | 0.258 | 220.059 | 0.000* | 0.022 | 0.013–0.036 |

Notes.

The following factors are significantly associated with odds of abnormal serum calcium –donation frequencies, age, and regions.

$n = 3{,}966$ in analysis of donation frequencies, gender, age, blood type and regions; $n = 3{,}567$ in analysis of BMI.

CI, confidence interval; OR, odds ratios.

*$p < 0.05$ by Binary logistic regression analysis.

(95% CI [1.489–2.398], $p < 0.001$), and 1.682 times (95% CI [1.309–2.163], $p < 0.001$) higher abnormal total serum protein odds as compared to donors aged 18-29. The odds of abnormal total serum protein in donors of Jiangxi, Hubei, Sichuan and Ningxia were decreased by 61.7% (95% CI [0.309–0.474], $p < 0.001$), 60.4% (95% CI [0.319–0.491], $p < 0.001$), 88.8% (95% CI [0.086–0.146], $p < 0.001$) and 91.9% (95% CI [0.061–0.108], $p < 0.001$) compared to Shandong participants, respectively (Table 5).

**Table 5  Binary logistic regression analysis between influencing factors and odds of abnormal total serum protein (lower than normal value range).**

| | B | SE (B) | Wald $\chi^2$ | $p$-value | OR | OR (95% CI) |
|---|---|---|---|---|---|---|
| **Donation frequencies** | | | | | | |
| 0 | 1.00 (referent) | | | | | |
| 1–6 | 1.251 | 0.116 | 115.487 | 0.000* | 3.494 | 2.781–4.390 |
| 7–11 | 0.861 | 0.119 | 52.795 | 0.000* | 2.366 | 1.875–2.984 |
| 12–27 | 0.779 | 0.120 | 41.805 | 0.000* | 2.179 | 1.721–2.760 |
| **Gender** | | | | | | |
| Male | 1.00 (referent) | | | | | |
| Female | −0.188 | 0.080 | 5.546 | 0.019* | 0.829 | 0.709–0.969 |
| **Age (years)** | | | | | | |
| 18–29 | 1.00 (referent) | | | | | |
| 30–39 | 0.362 | 0.129 | 7.865 | 0.005* | 1.436 | 1.115–1.850 |
| 40–49 | 0.636 | 0.122 | 27.357 | 0.000* | 1.889 | 1.489–2.398 |
| 50–60 | 0.520 | 0.128 | 16.499 | 0.000* | 1.682 | 1.309–2.163 |
| **Blood type** | | | | | | |
| A | 1.00 (referent) | | | | | |
| B | −0.017 | 0.101 | 0.029 | 0.864 | 0.983 | 0.806–1.198 |
| AB | −0.080 | 0.148 | 0.289 | 0.591 | 0.924 | 0.691–1.234 |
| O | −0.080 | 0.097 | 0.669 | 0.413 | 0.923 | 0.763–1.118 |
| **BMI** | | | | | | |
| <18.5 | 1.00 (referent) | | | | | |
| 18.5–23.9 | 0.485 | 0.272 | 3.178 | 0.075 | 1.624 | 0.953–2.767 |
| 24.0–27.9 | 0.440 | 0.272 | 2.625 | 0.105 | 1.553 | 0.912–2.647 |
| ≥28 | 0.408 | 0.278 | 2.151 | 0.142 | 1.503 | 0.872–2.593 |
| **Regions** | | | | | | |
| Shandong | 1.00 (referent) | | | | | |
| Jiangxi | −0.961 | 0.110 | 79.916 | 0.000* | 0.383 | 0.309–0.474 |
| Hubei | −0.927 | 0.110 | 70.917 | 0.000* | 0.396 | 0.319–0.491 |
| Sichuan | −2.186 | 0.134 | 266.733 | 0.000* | 0.112 | 0.086–0.146 |
| Ningxia | −2.516 | 0.147 | 294.796 | 0.000* | 0.081 | 0.061–0.108 |

Notes.

The following factors are significantly associated with odds of abnormal serum calcium –donation frequencies, age, and regions.

$n = 3,966$ in analysis of donation frequencies, gender, age, blood type and regions; $n = 3,567$ in analysis of BMI.

CI, confidence interval; OR, odds ratios.

*$p < 0.05$ by Binary logistic regression analysis.

## DISCUSSION

SP donors provide the starting material for manufacture of PDMPs. In China, SP is collected through plasmapheresis. The donation frequency of plasmapheresis donation is higher than whole-blood donation. Detailedly, the interval between two donations shall not be less than 6 months for whole-blood donation and shall not be less than 14 days for plasma donation (*Commission GOoNH, 2021*; *Congress CotNPs, 1997*). Donor safety has been a major concern throughout plasmapheresis donation. In this study, we evaluated the

influences of donation frequencies, blood type, gender, age, regions, and BMI on serum calcium and total serum protein in the Chinese plasma donation population.

Citrate was used as anticoagulant during apheresis donations to prevent coagulation and clotting in the apheresis circuit (*Grau et al., 2017*). As citrate is a chelator of ionized calcium, infusing sodium citrate will reduce blood level of ionized calcium (*Page & Harrison, 2012*). When plasma ionized calcium below 0.9 mmol/L, clinical symptoms of hypocalcemia and hypotension appeared (*Monchi, 2017*). It was reported that about 1% of first donations and about 0.3% of repeat donations had moderate to severe citrate effects (*McLeod et al., 1998*). Furthermore, acute perturbations in calcium metabolism followed apheresis donation (*Amrein et al., 2012*; *McLeod et al., 1998*; *Makar et al., 2002*). It is well known that serum calcium exists in three forms: ionized, protein-bound, and complexed. Among them, the normal value of ionized calcium is about half of the total serum calcium (*Tinawi, 2021*). Our study showed age, gender, donation frequencies, regions and BMI had influences on serum calcium level. Moreover, serum calcium level was lower than normal value range in 20.55% (815/3966) of the plasma donors, and donation frequencies, age, BMI and regions were significantly associated with a higher risk of low serum calcium level. Before discussing these associations, it must be pointed out that the serum calcium values of this study had not been adjusted to the albumin (or total protein) level by an appropriate formula, for the reason that many researchers have found the calcium detection value corrected by the formula cannot accurately reflect the calcium level, especially for people with abnormal calcium levels (*Ladenson, Lewis & Boyd, 1978*; *Law et al., 2021*; *Morton, Garl & Holden, 2010*; *Ohbal et al., 2014*; *Pfitzenmeyer et al., 2007*; *Smith, Wilson & Schneider, 2018*). Our results should be interpreted with caution.

The effects of age, sex and BMI on serum calcium levels have been extensively studied in general, healthy population (*Jafari-Giv et al., 2019*; *Zhang et al., 2014*; *Jorde, Sundsfjord & Bønaa, 2001*; *Rudnicki et al., 1993*; *Endres et al., 1987*; *Sokoll & Dawson-Hughes, 1989*; *Aoki, 1975*; *Sangal & Beevers, 1982*; *Jorde, Bonaa & Sundsfjord, 1999*), but few integrated reports in plasma donor population. In the present study, we have found serum calcium levels significantly associated with age. Similar results have been confirmed in general population (*Jorde, Sundsfjord & Bønaa, 2001*; *Sokoll & Dawson-Hughes, 1989*; *Aoki, 1975*; *Sangal & Beevers, 1982*). Conversely, some other studies have illustrated no differences in serum calcium levels and age (*Jafari-Giv et al., 2019*; *Endres et al., 1987*; *Jorde, Bonaa & Sundsfjord, 1999*). In addition, *Sweegers et al. (2021)* also found age was significantly associated with multiple post-donation symptoms (such as fatigue, dizziness, headache and higher energy level) in whole blood donors. Conflicting results also have been reported with respect to the effect of gender on serum calcium levels. *Jafari-Giv et al. (2019)* showed that the lower level of serum calcium in males than females. However, consistent with our results, *Jorde, Bonaa & Sundsfjord (1999)* and *Zhang et al. (2014)* reported that serum calcium levels are significantly higher in males than in females. Furthermore, some other studies have demonstrated no differences in serum calcium levels between male and female (*Rudnicki et al., 1993*; *Endres et al., 1987*). These discrepancies may be the results of menopausal status of female samples and hormonal disorders on Ca metabolism. BMI of subjects has been reported as another one of the influencing factors for serum calcium

**Table 6  Serum calcium (mmol/L) in relation to age and gender in different BMI (kg/m²) donors (n = 3567)[*].**

| | BMI | | | | | | | |
| | 18.5 | | 18.5–23.9 | | 24.0–27.9 | | ≥28 | |
| | No. | Serum calcium | No. | Serum calcium | No. | Serum calcium | No. | Serum calcium |
|---|---|---|---|---|---|---|---|---|
| *Age* | | | | | | | | |
| 18–29 | 59 | 2.40 ± 0.13 | 360 | 2.36 ± 0.14 | 223 | 2.36 ± 0.14 | 125 | 2.32 ± 0.17 |
| 30–39 | 10 | 2.33 ± 0.18 | 275 | 2.29 ± 0.14 | 297 | 2.29 ± 0.14 | 166 | 2.28 ± 0.16 |
| 40–49 | 12 | 2.29 ± 0.13 | 400 | 2.26 ± 0.14 | 434 | 2.26 ± 0.14 | 242 | 2.28 ± 0.14 |
| 50–60 | 5 | 2.31 ± 0.17 | 326 | 2.28 ± 0.13 | 450 | 2.28 ± 0.13 | 183 | 2.29 ± 0.14 |
| *p* | | 0.055 | | 0.000 | | 0.000 | | 0.078 |
| *Gender* | | | | | | | | |
| Male | 58 | 2.39 ± 0.14 | 641 | 2.39 ± 0.14 | 714 | 2.31 ± 0.14 | 374 | 2.30 ± 0.15 |
| Female | 28 | 2.34 ± 0.14 | 720 | 2.34 ± 0.14 | 690 | 2.34 ± 0.15 | 342 | 2.27 ± 0.16 |
| *p* | | 0.191 | | 0.011 | | 0.000 | | 0.001 |

**Notes.**
[*]The effects of gender on serum calcium was accomplished by using two-tailed unpaired Student's *t*-tests; multi-group comparisons were conducted by one-way ANOVA test., $p < 0.05$ was considered statistically significant.

levels (*Jafari-Giv et al., 2019*; *Jorde, Sundsfjord & Bønaa, 2001*). *Jorde, Sundsfjord & Bønaa (2001)* found that there was a significant positive association between serum calcium and BMI. On the contrary, *Jafari-Giv et al. (2019)* reported that obese subjects had a lower level of calcium, which was similar to our results. We also found age and gender had significant influence on serum calcium levels in the groups with a BMI of 18.5–23.9 and 24.0–27.9 (Table 6), but the mechanism for this still unknown.

Plasmapheresis donation is a process that takes only plasma from the donor while the cellular components are returned. Except water, plasma contains salts, enzymes, antibodies and other proteins. Plasma proteins, such as albumin, immunoglobulins, and coagulation factors, are the main source of PDMPs. Moreover, they are closely related to the immune system and maintain intravascular colloid osmotic pressure, transportation of various metabolites, and regulation of numerous physiological functions (*Chow, Fox & Gama, 2008*; *Malik et al., 2011*; *Neel, McClave & Martindale, 2011*; *Prajapati, Sharma & Roy, 2011*). It was reported that high-intensity plasmapheresis results in depletion of serum proteins with a long plasma half-life (*Amrein et al., 2012*). The loss of plasma proteins are associated with hypoproteinemia, hypoalbuminemia, hypogammaglobulinemia, and so on *Lundsgaard-Hansen (1980)*. In present study, we found the total serum protein levels of 27.99% (1,111/3,969) donors were lower than normal value range (65–85 g/L), and donation frequencies, gender, age and regions were significant determinants factors of the odds of abnormal total serum protein. Some researchers reported that the total serum protein levels of plasmapheresis donors were significantly lower than that of non-donor controls (*Bechtloff et al., 2005*; *Tran-Mi et al., 2004*), and plasma protein levels were significant different in plasmas collected with different frequencies (*Laub et al., 2010*), these generally accorded with our findings. Moreover, *Tian et al. (2014)* observed the total serum protein levels decreased with increasing age in elderly Chinese. Similar to this result, we found age was negative correlated with total serum protein levels (r = −0.120, $p < 0.001$)

**Table 7  Total serum protein (g/L) in relation to age and gender in different BMI (kg/m²) donors ($n = 3570$)[*].**

| | BMI | | | | | | | |
| --- | --- | --- | --- | --- | --- | --- | --- | --- |
| | <18.5 | | 18.5–23.9 | | 24.0–27.9 | | ≥28 | |
| | No. | Total serum protein | No. | Total serum protein | No. | Total serum protein | No. | Total serum protein |
| *Age* | | | | | | | | |
| 18–29 | 58 | 70.83 ± 5.81 | 361 | 69.71 ± 5.84 | 224 | 68.39 ± 7.12 | 125 | 68.81 ± 6.92 |
| 30–39 | 10 | 66.70 ± 6.27 | 275 | 67.59 ± 5.74 | 297 | 67.40 ± 6.23 | 166 | 66.85 ± 6.28 |
| 40–49 | 12 | 68.58 ± 6.36 | 401 | 66.71 ± 6.22 | 435 | 67.06 ± 5.97 | 242 | 67.48 ± 5.70 |
| 50–60 | 5 | 68.10 ± 7.04 | 326 | 66.28 ± 5.92 | 450 | 67.46 ± 5.58 | 183 | 67.15 ± 5.76 |
| *p* | | 0.162 | | 0.000 | | 0.069 | | 0.041 |
| *Gender* | | | | | | | | |
| Male | 58 | 69.72 ± 6.04 | 642 | 67.02 ± 6.25 | 716 | 67.48 ± 5.91 | 374 | 67.55 ± 5.92 |
| Female | 27 | 70.20 ± 6.29 | 721 | 68.08 ± 5.92 | 690 | 67.46 ± 6.33 | 342 | 67.42 ± 6.30 |
| *p* | | 0.735 | | 0.001 | | 0.945 | | 0.777 |

**Notes.**

[*]The effects of gender on serum calcium was accomplished by using two-tailed unpaired Student's *t*-tests; multi-group comparisons were conducted by one-way ANOVA test., $p < 0.05$ was considered statistically significant.

in plasma donors. Additionally, we confirmed that the total serum protein level of plasma donors also been influenced by gender and BMI. Age and gender had significant influence on serum calcium levels in the group with a BMI of 18.5–23.9 (Table 7). The mechanism for this result needs further research.

In addition, we found donors coming from Shangdong province were highly represented in abnormal calcium and protein amount groups. Shangdong is the only coastal province among the five provinces participating in this research, the different lifestyle and environmental conditions may affecting serum calcium and serum total protein levels. The reasons of this result need further study.

There were some limitations to this study. Firstly, in our study we were limited to investigate the association between six influencing factors and levels of serum calcium and total serum protein. Other influencing factors, such as menopause, circulating blood volume or ethnicity may also affect levels of serum calcium and total serum protein. Secondly, in this study, all the participants were recruited from five provinces of China, and it is unclear whether the results can be generalized and are applicable to plasma donors in other provinces in China or other countries. Future research in other locations will assess in greater detail the general applicability of our findings.

## CONCLUSION

In summary, our study clearly suggested that donation frequency, gender, age, regions, and BMI can influence levels of serum calcium and total serum protein. In plasma donors, older age, being repeat donors, BMI was 24.0–27.9, and come from Shandong province were significantly associated with higher odds of abnormal serum calcium. Furthermore, being repeat donors, older age, being male, and come from Shandong province were significantly correlated with higher odds of abnormal total serum protein. Accordingly, in order to

reduce the probability of low serum calcium and low total serum protein, more attention should be paid to the age, donation frequency and region of plasma donors. This study's findings provide new information on Chinese plasma donors and may contribute toward improving the safety of plasma donation.

## ACKNOWLEDGEMENTS

We would also like to thank the staff of the 10 plasmapheresis centers who all have contributed with the work of recuiting donors in this study. Furthermore, our thanks to all the donors participating in our study.

### Funding
This work was supported by CAMS Innovation Fund for Medical Sciences [grant number 2021-I2M-1-060]. The funders had no role in study design, data collection and analysis, decision to publish, or preparation of the manuscript.

### Grant Disclosures
The following grant information was disclosed by the authors:
CAMS Innovation Fund for Medical Sciences: 2021-I2M-1-060.

### Competing Interests
Demei Dong and Yang Gao are employees of Beijing Tiantan Biological Products Co., LTD. Ding Yu is an employee of Chengdu Rongsheng Pharmaceuticals Co., LTD. The authors declare there are no competing interests.

### Author Contributions

- Bin Liu conceived and designed the experiments, performed the experiments, prepared figures and/or tables, authored or reviewed drafts of the article, and approved the final draft.
- Demei Dong conceived and designed the experiments, performed the experiments, prepared figures and/or tables, authored or reviewed drafts of the article, and approved the final draft.
- Zongkui Wang performed the experiments, prepared figures and/or tables, authored or reviewed drafts of the article, and approved the final draft.
- Yang Gao performed the experiments, prepared figures and/or tables, authored or reviewed drafts of the article, and approved the final draft.
- Ding Yu performed the experiments, prepared figures and/or tables, authored or reviewed drafts of the article, and approved the final draft.
- Shengliang Ye performed the experiments, analyzed the data, prepared figures and/or tables, authored or reviewed drafts of the article, and approved the final draft.
- Xi Du conceived and designed the experiments, performed the experiments, prepared figures and/or tables, authored or reviewed drafts of the article, and approved the final draft.

- Li Ma performed the experiments, prepared figures and/or tables, authored or reviewed drafts of the article, and approved the final draft.
- Haijun Cao conceived and designed the experiments, prepared figures and/or tables, authored or reviewed drafts of the article, and approved the final draft.
- Fengjuan Liu analyzed the data, prepared figures and/or tables, authored or reviewed drafts of the article, and approved the final draft.
- Rong Zhang conceived and designed the experiments, analyzed the data, prepared figures and/or tables, authored or reviewed drafts of the article, and approved the final draft.
- Changqing Li conceived and designed the experiments, prepared figures and/or tables, authored or reviewed drafts of the article, and approved the final draft.

## Data Availability

The raw data is available in the Supplemental File.

## Supplemental Information

Supplemental information for this article can be found online at http://dx.doi.org/10.7717/peerj.14474#supplemental-information.

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
