# Peer review of "Analysis of influencing factors of serum total protein and serum calcium content in plasma donors"

_PeerJ, doi:10.7717/peerj.14474_

## Round 0.1 · original submission · Major Revisions

The manuscript is relevant to the journal's scope and has a significant interest in the journal's readership. As stated by the reviewers, there are some major comments that need to be addressed. Please provide a rebuttal for each comment raised by the reviewer and make the necessary changes for resubmission.

·

Basic reporting

I had the pleasure of reviewing the manuscript titled “Analysis of influencing factors of serum total protein and serum calcium content in plasma donors”. This study elucidates factors like donation frequency, age, gender, BMI, and location which may lead to reduced serum calcium and serum protein levels in people donating plasma. The study findings are significantly relevant to the safety and well-being of plasma donors. There are however a few comments and suggestions that may further enhance the robustness and readability of the manuscript which is already succinctly written.

1. Sentences highlighted in yellow in the annotated manuscript need grammatical proofreading and editing.
2. The order of figures should match the order of the text or vice versa for better readability.
3. The authors state that the data is represented as the mean ± SD; however, the corresponding figures are represented with medians. Could the authors please make the text and figures consistent in terms of the central tendency descriptor for the data? A combination of mean + SD along with data spread for the figures would enhance the clarity.
4. Lines 62-64 need references.
5. Lines 104-105 – Please specify what classifies as abnormal serum calcium and total protein in the methods or results section as well (apart from the table caption).
6. Could the authors please crosscheck and provide more information for finding the references?
7. Lines 212 -215. The authors mention a similarity between the post-donation symptoms in Sweeger’s study and low calcium found in this study as an effect of age. Could the authors please specify what post-donation symptoms from Sweeger’s study correlate with low calcium found in their study?
8. Please add the limitations of this study in the discussion.

Experimental design

9. Normal serum calcium levels are usually dependent on age and gender. Was the number of people with abnormal levels normalized based on age and gender or a single value was used as demarcation? Also, was menopause considered as a potential factor affecting these values? The manuscript would benefit from the addition of these discussion points.
10. Please also consider the intersectionality between age and BMI and Gender and BMI to make recommendations and how those factors affect the number of donations. Such information may hold a significant impact on minimizing the negative effects of plasma donation.
11. Donors from Shandong province seem to be highly represented in abnormal calcium and protein amount groups. Could the authors please comment on the potential cause of such a trend? Is such an observation influenced by some technicality at the plasma donation center or a lifestyle trend in that region?

Validity of the findings

no comment

Additional comments

no comment

Reviewer 2 ·

Basic reporting

acceptable

Experimental design

The article is well written, however, several ambiguities need to be clarified in the text, especially regarding the method section.
1- Basic Reporting: acceptable
2- Experimental design
My comments are as follows:
Abstract:
1- The methods section needs to be more developed
2- There is a biochemical discrepancy; serum calcium and serum protein, as their names indicate, are measured in serum and not in plasma; the authors say that serum calcium and serum protein were measured in donors' plasma, which is a biochemical and analytical discrepancy. Since the assay was performed on serum, the word "plasma" should be replaced by serum.
Introduction:
1- Line 53, 61: a phrase never begins with "and", the dot should be replaced by a comma.
Methods:
1- The study design was not stated, in this case it is a cross-sectional, multicenter study.
2- Calcium status is mainly related to vitamin D and PTH status, exclusion criteria should also include any disturbance of these two parameters, in particular severe vitamin D deficiency and hyperparathyroidism. Similarly, patients with a chronic inflammatory syndrome should also be excluded as this condition leads to elevated plasma protein levels
3- The assay techniques for serum calcium and serum protein must be indicated, not just the reagent brand.
4- The reference ranges of these two parameters must be mentioned in the methods section.
5- Since most of the plasma calcium is bound to serum albumin, total blood calcium should be adjusted to the albumin (or total protein) level by an appropriate formula.
Results:
1- Line 115-119: indicate the table number in which these results are shown.
2- The mean values of blood calcium and blood protein do not appear anywhere in the tables and figures, what is shown in figures 1 and 2 represents the median and not the mean.
3- Line 130 and 142: There is a biochemical discrepancy, men had significantly higher levels of plasma calcium and lower levels of protein, compared to women. It is known that in healthy subjects a positive correlation is found between protein and total calcium levels, so how to explain this discrepancy?
4- The authors should indicate the normal BMI values for the Asian population, because for the "non-Asian" populations, a BMI between 24-27 kg/m2, is considered normal or simply overweight and not obesity.
Discussion:
1- The discussion needs to be improved, in deed, most of this discussion is seen as a repetition of the "results" section, the authors need to give more explanation about the mechanisms of these disturbances, and should compare their results with previous studies.
2- The authors should add a section that discusses the strength and limitations of this study.
3. Validity of the Findings: acceptable.

Validity of the findings

acceptable

Additional comments

/

---

## Round 0.2 · accepted · Accept

Thank you for addressing all the comments that the reviewers raised. All the reviewers for this manuscript were satisfied with the revised resubmission. The authors are requested to give the manuscript one final read-thru to ensure that any typographical and grammatical errors are fixed.

·

Basic reporting

The Mauscript requires thorough proofreading for grammatical accuracy.

Experimental design

N/A

Validity of the findings

N/A

Additional comments

Rest everything looks good!

Reviewer 2 ·

Basic reporting

Accepted

Experimental design

I would like to thank the authors for responding to most of the questions and suggestions. I would still like to point out the references listed by the author, which doubt the validity of the correction of blood calcium levels by an appropriate formula; these references concern patients with chronic end-stage renal disease, dialysis patients, hospitalized patients and very old patients, in these patients the causes of calcium balance disturbance are very numerous, and are not limited to the decrease of albuminemia, in the case of this study, where plasmapheresis could cause hypoalbuminemia, the correction of the blood calcium level is more than useful in order to verify the independence of the association with the decrease of the blood calcium level

Validity of the findings

Accepted